# Assessing the Speciation of *Lutjanus campechanus* and *Lutjanus purpureus* through Otolith Shape and Genetic Analyses

Angel Marval-Rodríguez [1], Ximena Renán [2,*], Gabriela Galindo-Cortes [1], Saraí Acuña-Ramírez [1], María de Lourdes Jiménez-Badillo [1], Hectorina Rodulfo [3], Jorge L. Montero-Muñoz [2], Thierry Brulé [2] and Marcos De Donato [3,*]

1   Instituto de Ciencias Marinas y Pesquerías, Universidad Veracruzana, Boca del Rio 94290, Mexico; avgelo7@gmail.com (A.M.-R.); gagalindo@uv.mx (G.G.-C.); acua.sarai@gmail.com (S.A.-R.); ljimenez@uv.mx (M.d.L.J.-B.)

2   Departamento de Recursos del Mar, Centro de Investigación y de Estudios Avanzados del Instituto Politécnico Nacional, Mérida 97205, Mexico; jorge.montero@cinvestav.mx (J.L.M.-M.); tbrule@cinvestav.mx (T.B.)

3   Tecnológico de Monterrey, Escuela de Ingeniería y Ciencias, Querétaro 76130, Mexico; herodulfo@tec.mx

*   Correspondence: ximena.renan@cinvestav.mx (X.R.); mdedonate@tec.mx (M.D.D.); Tel.: +52-999-242-1587 (X.R.); +52-442-231-2927 (M.D.D.)

**Abstract:** Based on their morphological and genetic similarity, several studies have proposed that *Lutjanus campechanus* and *Lutjanus purpureus* are the same species, but there is no confirmed consensus yet. A population-based study concerning otolith shape and genetic analyses was used to evaluate if *L. campechanus* and *L. purpureus* are the same species. Samples were collected from populations in the southwestern Gulf of Mexico and the Venezuelan Caribbean. Otolith shape was evaluated by traditional and outline-based geometric morphometrics. Genetic characterization was performed by sequencing the mtDNA control region and intron 8 of the nuclear gene FASD2. The otolith shape analysis did not indicate differences between species. A nested PERMANOVA identified differences in otolith shape for the nested population factor (fishing area) in morphometrics and shape indexes ($p = 0.001$) and otolith contour (WLT4 anterior zone, $p = 0.005$ and WLT4 posterodorsal zone, $p = 0.002$). An AMOVA found the genetic variation between geographic regions to be 10%, while intrapopulation variation was 90%. Network analysis identified an important connection between haplotypes from different regions. A phylogenetic analysis identified a monophyletic group formed by *L. campechanus* and *L. purpureus*, suggesting insufficient evolutionary distances between them. Both otolith shape and molecular analyses identified differences, not between the *L. campechanus* and *L. purpureus* species, but among their populations, suggesting that western Atlantic red snappers are experiencing a speciation process.

**Keywords:** red snapper; sagittal otolith; wavelet; genetic diversity; population structure

## 1. Introduction

Correct species identification and data on population spatial distribution are vital to improving fishery resource assessment and management [1–3]. Although it is practically impossible to analyze all the variables (morphometrics and/or genetics) characteristic of a species, variation in the individual phenotypes of organisms or their genotypes can be examined to identify species and delimit populations [4,5]. Traditionally, fish are identified using molecular genetics, morphometric measurements and meristic characters, and otolith shape analysis, among other techniques [6]. Sequencing of conserved genes (a molecular biology technique) is the most common tool currently used for species identification, in addition to the characterization of population structure and gene flow [7]. Additionally, otolith shape analysis has been in use for more than 20 years as an objective method for

species identification, fish stock discrimination, systemic and taxonomic studies, aging of fish, fish auditory neuroscience studies, and the study of the ecomorphological patterns of fish [8–16].

Particularly for the Lutjanidae family, otolith shape analysis has been used to identify environmental and genetic influences on otolith morphology, to age juvenile red snappers, discriminate between stocks, identify closely-related species, and to analyze the morphometric relationships between two species [17–21].

The Lutjanidae family comprises a large group of species distributed in tropical and subtropical marine ecosystems in the Atlantic, Pacific, and Indian Oceans [22]. In the western Atlantic, 6 genera and 18 species have been identified, of which 12 species belong to the genus *Lutjanus* [23]. *Lutjanus campechanus* and *L. purpureus* are the most important species captured in the western Atlantic, and fetch high market prices [24]. Populations of *L. campechanus* are distributed throughout the Gulf of Mexico, from the Yucatan Peninsula to Key West, and along the Atlantic coast of the United States to Massachusetts [25]. *Lutjanus purpureus* are distributed from the southern coast of Cuba and the Yucatan Peninsula throughout the Caribbean Sea, and from the north and northeast of South America to Pernambuco in Brazil, approximately [26].

*Lutjanus campechanus* and *L. purpureus* are remarkably similar in their life cycle, population parameters, and morphology [27]. Taxonomic identification of these species is difficult because of the similarities in their external morphology and the overlap in the characteristics commonly used to identify them, such as spines, hard and soft rays of the pectoral, dorsal and anal fins, lateral line scales, and gill rakers [28,29]. Based on these morphological similarities, Cervigón et al. [30] hypothesized the existence of a single species of red snapper in the western Atlantic Ocean; that is, that *L. campechanus* and *L. purpureus* are actually the same species, with morphological differences between populations in the western Atlantic. Based on genetic analysis, lack of phylogeographic structure, and intense intermingling between individuals, they suggested the probable existence of just one red snapper species throughout the western Atlantic [29,31]. However, a recent study [32] used molecular delimitation to discriminate *L. campechanus* and *L. purpureus* as distinct evolutionary units, although the groups did share a significant number of haplotypes, suggesting important gene flow between them. The objective of the present study was to elucidate the species-specific boundaries between *L. campechanus* and *L. purpureus* for the western Atlantic by combining, for the first time, otolith morphometrics and genetic analyses.

## 2. Materials and Methods

### 2.1. Sample Collection

Biological samples of *L. campechanus* and *L. purpureus* were collected between 2015 and 2017 from dead individuals caught by a commercial multi-species artisanal fleet (Veracruz and Tabasco State) and an industrial shrimp trawl fleet in the southwestern Gulf of Mexico (Campeche State) and the Venezuelan Caribbean (Nueva Esparta and Sucre State) (Figure 1). The individuals comprised 108 *L. campechanus* (72–452 mm total length (TL)) and 24 *L. purpureus* (214–460 mm TL). Sagittal otoliths were extracted through the gill arch, washed with distilled water and stored in labeled plastic containers. Collected muscle tissue was stored in 96% ethanol and kept frozen until laboratory processing.

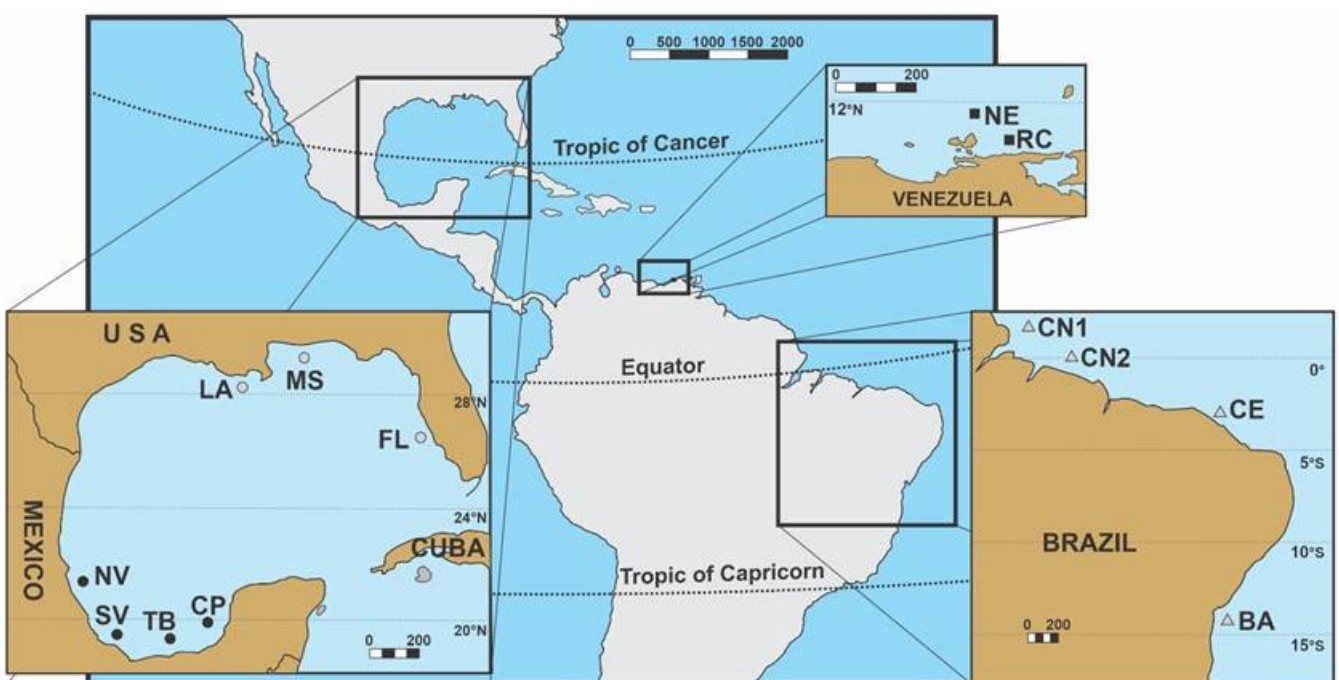

**Figure 1.** Map of *Lutjanus campechanus* and *Lutjanus purpureus* spatial distribution. Black point: fishing areas used in this study. Gray point: sequences from GenBank of samples collected in the USA and Brazil. (NV = North Veracruz; SV = South Veracruz; TB = Tabasco; CP = Campeche; NE = Nueva Esparta; RC = Rio Caribe; LA = Louisiana; MS = Mississippi, FL = Florida; CN1 = Coast North Brazil (46° and 50° W); CN2 = Coast North Brazil (43° and 45° W); CE = Ceará; BA = Bahia).

*2.2. Otolith Analysis*

2.2.1. Otolith Morphological Description

The morphology of the left otoliths from *L. campechanus* and *L. purpureus* were described based on high-definition photographs taken with a scanning electron microscope (FESEM-JOEL-7600F). Morphological description was based on thirteen characteristics considering general otolith shape, edges, sulcus acusticus, ostium, cauda, rostrum and antirostrum, and variations in shape, position and orientation [33,34].

2.2.2. Otolith Shape Analyses

Digital photographs of the left otoliths were taken using a stereoscopic microscope (Leica-EZ4E), with the sulcus acusticus facing downwards, under reflected light on a black background. Based on the morphometrics classification established by Pavlov [35], we used traditional morphometrics and outline-based geometric morphometrics to analyze otolith shape. Traditional otolith morphometrics, which describe shape and length measurements or indices between vectors passing through certain points, were automatically generated using the Image Pro Plus v.7 software (Media Cybernetics Inc., USA). Four otolith morphometrics were measured (area (A); perimeter (P); maximum diameter (MaxD); and minimum diameter (MinD)), as well as five shape indices (aspect (AS); ellipticity (E); rectangularity (RE); roundness (RD) and fractal dimension index (FI)).

The outline-based geometric otolith morphometrics analysis was performed by extracting discrete Wavelet Transforms (DWLT) using the "Shape" module in the Age & Shape program (Infaimon, Spain). This program reconstructs otolith contour by tracing equally sampled angles (radii) from the otolith geometrical center (mean x and y polar coordinates) to 512 equidistant points in the edge. It then automatically generates ten multiscale decompositions from the finest (DWLT1) to the coarsest (DWLT10) [36–38]. Following Tuset et al. [34], the differential characteristics of each otolith, by subsections (anterior, ventral, posterodorsal and anterodorsal), were defined by the graphic representation of wavelet scale mean and standard deviation.

2.2.3. Otolith Statistical Analysis

Fish size effects were removed from the magnitude of otolith morphometrics and shape indices, as recommended [39,40]. To remove amplitude effects, all DWLT multi-scale decompositions were standardized by dividing each radius by the mean radius length [41]. A principal component analysis (PCA) was applied to reduce the dimensions without loss of information, selecting the DWLT scale with the strongest correlation to component 1 [42]. This analysis indicated that the DWLT4 scale explained 94% of data variability; it was used in the subsequent analyses.

Six steps were applied in the statistical analysis to estimate differences in otolith shape in relation to the population factor and region factor. First, morphometrics, shape indices and DWLT4 data were transformed using the Hellinger distance. Second, a Euclidean triangular matrix was calculated using the Euclidean distance function applied to the Hellinger-transformed data [43]. Third, the resulting Hellinger distance matrix was used in a multivariate permutational analysis of variance (PERMANOVA) [44] to assess the region factor and the nested factor of population. Fourth, for the significant factor in the PERMANOVA ($\alpha = 0.05$), pairwise tests were used to estimate differences in otolith shape between factor levels. Fifth, the significant factor was plotted using the multivariate of the metric dimensional scale (MDS) analysis. Sixth and final, an average of bootstrap samples (95% confidence bootstrap region) was generated for the MDS to compare factor levels. The minimum dimension value of the MDS metric was 0.99 Pearson's correlation. The triangular Hellinger distance matrix, PERMANOVA and MDS analyses were run with PERMANOVA + for PRIMER (FIRST-E: Plymouth, UK) [45].

*2.3. Molecular Analysis*

2.3.1. Sample Processing, Fragment Amplification and Sequencing

DNA was extracted from alcohol-preserved muscle tissue using the Wizard® Genomic DNA Purification Kit (Promega, Madison, USA), following the manufacturer protocol. DNA concentration and quality were measured using a NanoDrop One (Thermo Scientific, Waltham, USA) and by visual inspection of the DNA by electrophoresis. DNA extracts were stored at $-20$ °C until use.

Genetic characterization of the samples was performed by sequencing the D-loop or control region of the mitochondrial DNA (mtDNA-CR) and the intron 8 of the nuclear gene for the enzyme fatty acid desaturase 2 (FADS2), according to previously published studies [46,47]. The fragments were sequenced using both primers and the BigDye Terminator system at the National Biodiversity Genomic Laboratory (Laboratorio Nacional de Genómica para la Biodiversidad—Langebio), Irapuato, Mexico.

2.3.2. Population Analysis

Evaluation of population structure was performed by calculating the number of haplotypes (h) and nucleotide diversity (π) within and between the populations (fishing areas) using the Arlequin v.3.5 software [48]. Sequence population structure was assessed using the fixation index (FST), and an analysis of molecular variance (AMOVA). Haplotype relationships were reconstructed using the Network ver. 10.1 software (Fluxus Technology, Ltd., Santa Clara, USA), and calculated with the median joining algorithm [49] using default settings (weight: 10 and ε: 0). A test to identify the correlation between genetic and geographic distances between populations was performed with a Mantel test using the geographic distance calculated from Google Earth (i.e., straight lines between sampling areas) and pairwise FST (obtained from the AMOVA). This was run with the GenAlEx v.6.51b2 software [50].

2.3.3. Phylogenetic Analysis

The generated sequences were viewed using the GeneStudio v.2.2.0 program (http://genestudio.com, accessed on 14 Mach 2022) to assess sequence quality and identify differences between them. They were compared to sequences in GenBank using BLAST.

Phylogenetic analyses were run using a Bayesian phylogenetic analysis performed with the Mr.Bayes v.3.2.1 program [51], implementing the general time-reversible (GTR) model using the rate at each site as a random variable. A discrete gamma distribution was applied to model evolutionary rate differences between sites (G) and a proportion of invariable sites. Markov chain Monte Carlo (MCMC) chains were run for 1,000,000 generations.

The sequences generated in the present study were deposited in GenBank (Table S1). Sequences from GenBank from samples collected in the USA and Brazil for the mtDNA-CR region, and from Brazil for the FADS2 gene, were used in the present study (Table S2). To evaluate the phylogeographic attributes of *L. campechanus* and *L. purpureus*, phylogenetic analysis consensus sequences were used for three *Lutjanus* species (Table S3) distributed along the Pacific and Atlantic coasts of the Americas: *L. peru*, *L. synagris* and *L. guttatus*.

## 3. Results

### 3.1. Otolith Analysis

#### 3.1.1. Otolith Morphological Description

A total of 132 otoliths were analyzed (108 from *L. campechanus* and 24 from *L. purpureus*). All exhibited a generally pentagonal-like shape with a concave-convex profile. The sulcus acusticus was heterosucoidal, ostial with a middle position and downward orientation. Both species exhibited a developed rostrum and a moderately curved cauda (Figure 2). Despite their morphological similarities, the otoliths did vary, primarily in terms of the anterior and posterodorsal regions, ostium shape and margin type. Those from individuals from the Gulf of Mexico had an angled anterior region, an oblique posterodorsal region, and a poorly developed antirostrum. Also present were a sinuous ventral edge and angular dorsal edge with a funnel-liked ostium. In contrast, otoliths from individuals from the Venezuelan Caribbean exhibited a rounded anterior region and angular posterodorsal region, with crenate dorsal and ventral margins, a developed antirostrum and a rectangular ostium.

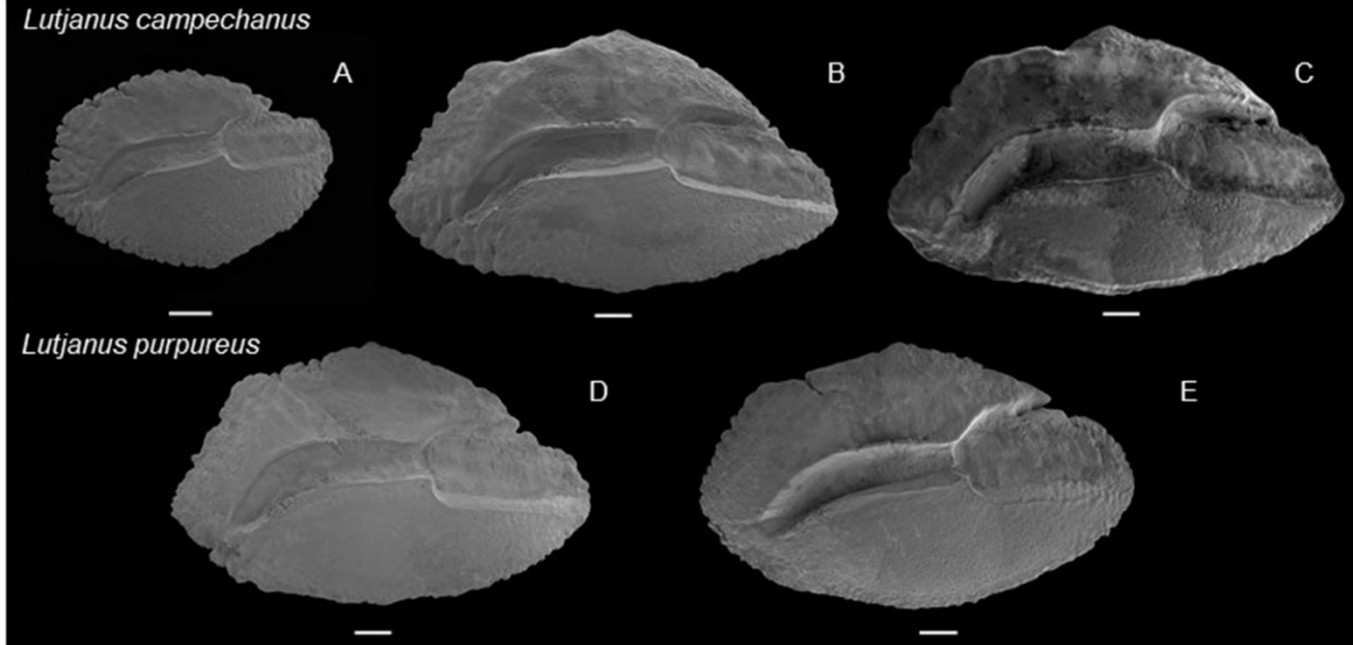

**Figure 2.** Red snapper sagittae otoliths: Gulf of Mexico (**A** = Campeche: 114 mm total length (TL); **B** = Tabasco: 360 mm TL; **C** = Veracruz: 378 mm TL) and Venezuelan Caribbean (**D** = Sucre: 285 mm TL; **E** = Nueva Esparta: 299 mm TL). Scale = 1 mm.

### 3.1.2. Inter-region Factor Variation

The nested PERMANOVA results using the data from the two regions (Gulf of Mexico and Caribbean basin) exhibited no differences in the morphometrics and shape indices ($p = 0.116$), DWLT4 anterior zone ($p = 0.967$), or DWLT4 posterodorsal zone ($p = 0.475$).

### 3.1.3. Intrapopulation Factor Variation

The nested PERMANOVA analysis identified differences between the otoliths from organisms caught in the different populations: morphometrics and shape indices ($p = 0.001$), DWLT4 anterior zone ($p = 0.005$), and DWLT4 posterodorsal zone ($p = 0.002$). The MDS analyses found that, although there was high variability in otolith shape between the individuals from different populations, only the individuals from Campeche (Gulf of Mexico) could be discriminated from other populations from the Gulf of Mexico and the Caribbean basins (Figure 3).

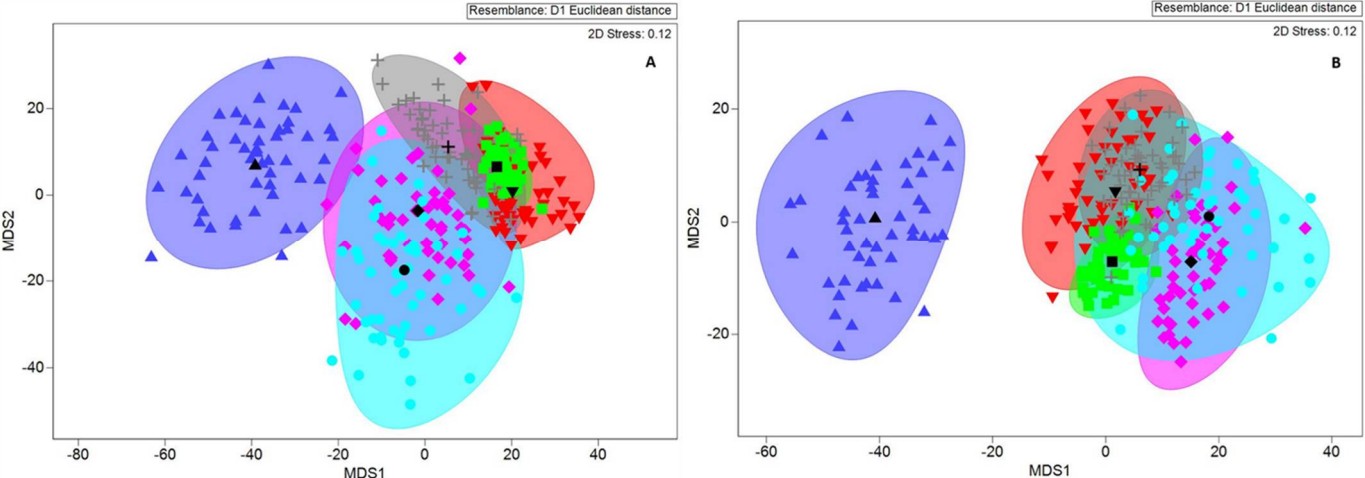

**Figure 3.** MDS arrangement diagram of Euclidean distances from the nested factor population in PERMANOVA of the DWLT4 anterior region (**A**) and DWLT4 posterior region (**B**) of sagittae otoliths from western Atlantic red snappers. (Blue = Campeche; Red = South Veracruz; Green = North Veracruz; Pink = Tabasco; Gray = Rio Caribe; Light blue = Nueva Esparta).

A pairwise comparison found differences between otoliths from individuals from Campeche vs. Tabasco ($p = 0.005$) and North Veracruz ($p = 0.010$) in the morphometrics/shape indices and anterior subsection. There were also differences in the DWLT4 posterodorsal subsection in otoliths from Campeche and Tabasco versus all other Gulf of Mexico populations ($p = 0.001$). Differences were also present in North Veracruz vs. Nueva Esparta ($p = 0.025$) for the morphometrics/shape indices and DWLT4 anterior subsection ($p = 0.023$), and in the DWLT4 posterodorsal otolith subsection in Campeche and Tabasco vs. Nueva Esparta and Rio Caribe ($p = 0.001$).

### 3.2. Molecular Analysis

### 3.2.1. Intrapopulation Factor Variation

A total of 1363 nucleotide positions were analyzed in the final dataset, 798 nucleotides for the mtDNA-CR and 565 nucleotides for the nuclear gene FADS2. For the mtDNA-CR region, significant intrapopulation values were found for nucleotide diversity, ranging from 0.013–0.029 substitutions per site. The populations from Venezuela and Brazil exhibited the highest nucleotide diversity values. The Tajima D test for neutrality for the mtDNA-CR region revealed that the changes were significantly different from random changes (non-neutral, $p < 0.05$) among regions of the USA and Mexico, but not within any of their populations. In Brazil, there were non-neutral changes between populations, and between North Coast individuals (Table S4).

Intrapopulational nucleotide diversity for the FADS2 gene was significantly lower, ranging from 0.002–0.012 substitutions per site. The Tajima test revealed that changes found within South Veracruz were non-neutral. It also identified non-neutral changes between the Venezuela populations and between Rio Caribe and Nueva Esparta individuals (Table S5), suggesting that selection or other forces are directly driving variability.

The AMOVA found that for mtDNA-CR, 53% of variation was intrapopulational and 47.9% was attributed to variation among geographical regions. For intron 8 (FADS2), 90% of variation was intrapopulational, with only 18% between regions (Table 1).

**Table 1.** Analysis of molecular variance (AMOVA) between regions (Gulf of Mexico and Caribbean basins) and populations (fishing areas) of red snapper using mitochondrial DNA based on the D-loop region and nuclear DNA based on the intron 8 (FADS2) sequences. (DF = degrees of freedom; SS = sum of squares; VC = variance components).

| Gene | Source of Variation | DF | SS | VC | % Variation |
|---|---|---|---|---|---|
| mtDNA D-loop | Inter-regional | 4 | 1231.22 | 8.59 | 47.86 |
| | Interpopulational within regions | 8 | 57.24 | −0.17 | −0.94 |
| | Intrapopulational | 199 | 1896.17 | 9.53 | 53.08 |
| | Total | 211 | 3184.62 | 17.95 | |
| FADS2, intron 8 | Inter-regional | 2 | 25.60 | 0.49 | 18.21 |
| | Interpopulational within regions | 4 | 3.09 | −0.23 | −8.44 |
| | Intrapopulational | 99 | 241.61 | 2.44 | 90.23 |
| | Total | 105 | 270.30 | 2.70 | |

### 3.2.2. Interpopulation Factor Differentiation

The inter-region pairwise FST values calculated using the mtDNA-CR (Table 2), were significant between the Gulf of Mexico and the Caribbean, as well as between the Gulf of Mexico and Brazil, but not between the USA and Mexico, nor Venezuela and Brazil. Average calculated inter-region FST values were five times lower than the average values between any of the regions and *Lutjanus peru*.

**Table 2.** Pairwise $F_{ST}$ values (above the diagonal) and the associated *p* value (below the diagonal) from the inter-regional comparison using the mtDNA D-loop sequences. * Statistically significant.

| Regions | USA | Mexico | Venezuela | Brazil | *L. peru* |
|---|---|---|---|---|---|
| USA | _ | 0.050 | 0.255 | 0.144 | 0.751 |
| Mexico | 0.121 | _ | 0.273 | 0.155 | 0.765 |
| Venezuela | <0.001 * | <0.001 * | _ | 0.003 | 0.716 |
| Brazil | <0.001 * | <0.001 * | 0.387 | _ | 0.659 |
| *L. peru* | <0.001 * | <0.001 * | <0.001 * | <0.001 * | _ |

Inter-population pairwise FST values (Table S6) were significant between some of the USA and Mexican populations, and between all the Gulf of Mexico populations and the Venezuela and Brazil populations, but not between populations from the same region nor between the Venezuela and Brazil populations.

In the case of the FADS2 gene, significant FST values were found only between Mexico and Brazil, but not between these two regions and Venezuela. Similar results were observed in the inter-population FST values, in which only populations from Mexico and Brazil differed (Table S7).

The Mantel tests revealed a significant correlation between geographic and genetic distance between populations (R = 0.571; *p* = 0.010; Figure 4), suggesting minimal inter-regional gene flow.

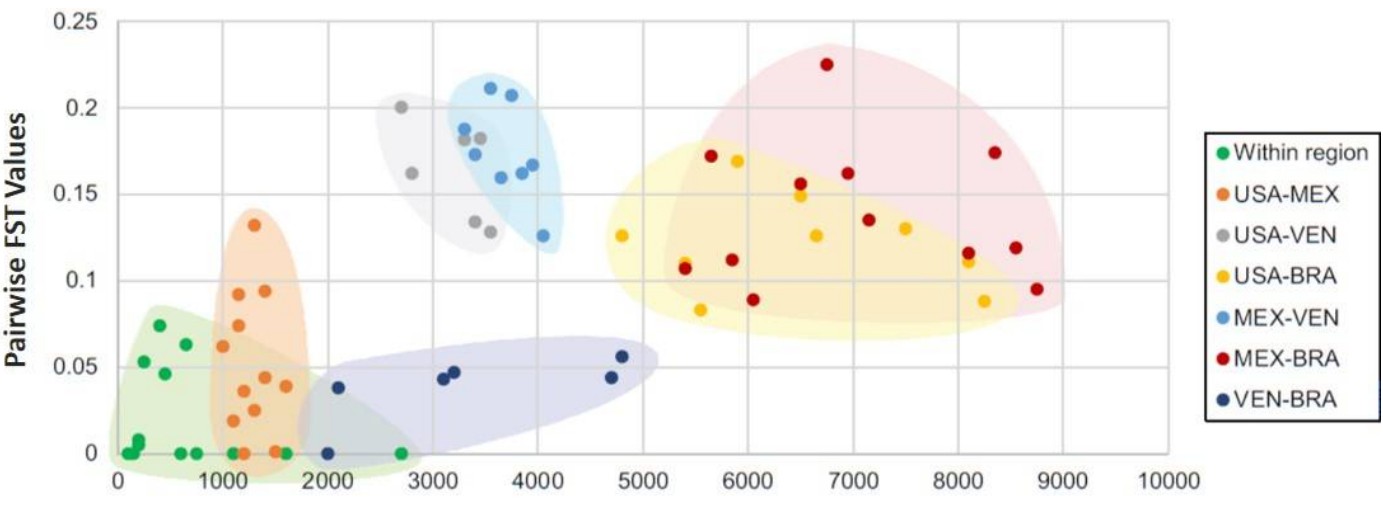

**Figure 4.** Relationship between geographic distance and linearized FST for western Atlantic red snapper.

3.2.3. Haplotype Network and Phylogenetic Analysis

The network analysis of the mtDNA-CR haplotypes (Figure 5A) showed wide variation in haplotypes. However, there was also substantial connection between haplotypes from different regions, with the most frequent haplotype being found in the USA, Mexico and Brazil. In many cases, the inter-regional difference between haplotypes was 1–6 mutations, as shown in the central area of the network. The haplotype network for the FADS2 gene (Figure 5B) exhibited a significantly lower level of variation compared to the mtDNA-CR region results, and the most frequent haplotypes were detected in all seven studied locations. The highest level of variation occurred in Brazil, and was most likely related to the total number of haplotypes from that region.

Phylogenetic analyses were performed using consensus sequences of all the sampling sites and the published mtDNA-CR region consensus sequences from *L. synagris* (same distribution as *L. campechanus*/*L. purpureus*), *L. guttatus* (Pacific coast of North and South America) and *L. peru* (Pacific coast of North and South America), used as an outgroup, demonstrated monophyly for individuals sampled in the Gulf of Mexico and for those sampled in the Caribbean and Brazil (Figure 6). Even though the statistical support was lower within each *L. campechanus* and *L. purpureus* population, and the relative evolutionary distance between them was small, statistical support still showed 100% incipient separation between these two groups. Of course, the distance between them was relatively much lower than the distances between them and the other *Lutjanus* species in the Americas.

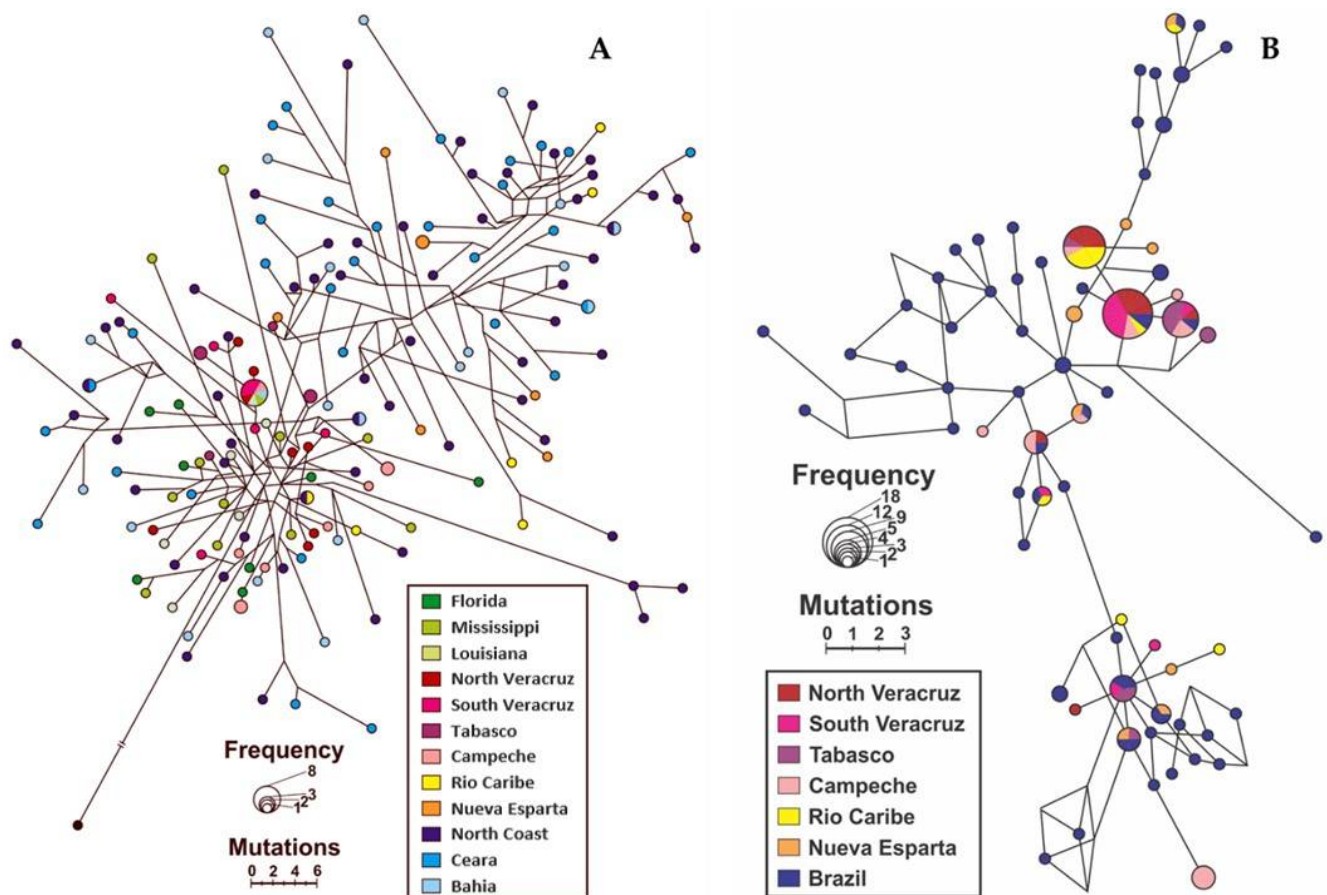

**Figure 5.** Median-joining network analysis for mtDNA-CR haplotypes (**A**) and FADS2 haplotypes (**B**) for *Lutjanus campechanus* and *Lutjanus purpureus* from the western Atlantic. Median vectors are not shown for clarity. The circles represent each haplotype, and circle size is proportional to haplotype frequency. Each color corresponds to a specific population and each circle shows a proportion of individuals in the haplotypes. Branch lengths are proportional to the number of S substitutions per nucleotide site. Mutational steps between haplotypes are represented by dashes.

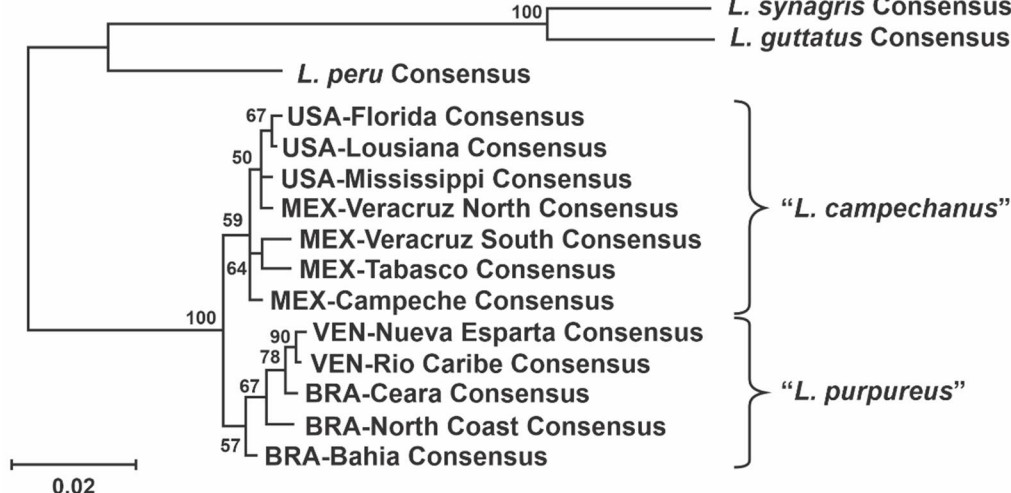

**Figure 6.** Phylogenetic analysis of the consensus sequences of the mtDNA region for *Lutjanus campechanus* and *Lutjanus purpureus* from the western Atlantic, and consensus sequences for other *Lutjanus* species from the Atlantic and Pacific coasts of the Americas.

## 4. Discussion

### 4.1. Otolith Analysis

Otolith morphologies in *L. campechanus* and *L. purpureus* were similar to that described for other species of the Lutjanidae family [25]. Only one description of otolith morphology in *L. campechanus* has been published to date [52], meaning the present study greatly expands the descriptions available for *L. campechanus*. It is also the first description of *L. purpureus* otoliths. The studied otoliths varied primarily in terms of the anterior and posterodorsal regions, ostium shape, and margin type. This variability has been noted in many types of fishes, especially in terms of otolith growth [15,20]. Several studies have verified that the anterior and posterodorsal regions and margin type are the most important in defining overall otolith shape, which is linked to ecological traits [15].

Otolith morphometric analysis cannot identify the main sources of variability. However, environmental conditions and genetic factors, or a combination thereof [53,54], can produce changes in fish growth rate and therefore in otolith shape, following an allometric increase in dimensions [55]. Interpopulation variation in otolith shape was noted in the otolith morphometrics/shape indices and anterior and posterior otolith contour subsections, which are all partially linked to different environmental factors for each area, and/or ontogenic factors [52]. The intrapopulation differences observed in otolith shape in individuals captured in Campeche is partly due to a combination of fish size and ontogenetic development. During early life stages, otoliths are still small, with a relatively high accretion rate, and can therefore be strongly influenced by environmental factors [56]. While otolith shape is genetically constrained, growth patterns in calcified structures can be affected by a wide range of exogenous factors, producing variation among conspecific individuals that have experienced contrasting life histories [57].

### 4.2. Genetic Differentiation

The molecular results from the mtDNA-CR and FADS2 intron showed that most of the genetic variation was intrapopulational. Nucleotide diversity values were low between the studied populations, indicating low genetic differentiation among red snapper populations from regions in the western Atlantic. The results also indicated an excess of polymorphisms, which is consistent with increasing population size and thus suggests that the studied populations are currently expanding. Gomes et al. [31], reported similar results when comparing the population structure of *L. campechanus* and *L. purpureus* in the western Atlantic. Considering the combined values of haplotype diversity and nucleotide diversity, Grant and Bowen [58] classified marine fish into four categories. Red snapper falls into category 1, that is, populations with low nucleotide diversity. This suggests that a population is expanding after a period of low effective population size, with rapid population growth based on one or a few lineages. This nucleotide diversity behavior is caused by a lack of physical barriers (terrestrial) and/or soft interregional barriers (nonterrestrial), facilitating the migration of adults and larvae and egg dispersal which promotes population growth or the establishment of new populations [59].

The present results indicate a generalized distribution of many haplotypes among the studied western Atlantic red snapper populations. The mixture of current and historical haplotypes by gene flow between different locations probably contributed to this result. Grant and Bowen [58] suggest that high levels of haplotype diversity indicate a long, stable evolutionary history or secondary contact between differentiated lineages. Genetic connectivity via ocean currents has been reported in different species of lutjanids [60] and serranids [61] in the western Atlantic.

The connection between haplotypes from populations in different regions of the western Atlantic shows the presence of gene flow (past and/or present), and suggests that small amounts of gene flow may be enough to homogenize red snapper populations, even in the face of demographic discontinuity. Although the analyzed populations are spatially separated, they may have had sufficient contact in the recent past to allow enough gene flow for haplotypes to spread into different geographical areas [58]. The phylogeny

inferred from mtDNA-CR in the studied western Atlantic red snapper populations clearly showed a shared common evolutionary history, while the gene flow reflected an incomplete separation of lineages with the retention of an ancestral polymorphism. This would explain the absence of simple monophyly between the two species and, given that they still share many mitochondrial haplotypes, may indicate that the cladogenetic event that gave rise to the two groups was relatively recent. A second alternative is the possibility of hybridization or introgression resulting from the generation of a fertile hybrid. This would then reproduce with members of one or both original species creating gene flow, as suggested by Pedraza-Marron et al. [62].

Gomes et al. [29] were the first to use molecular data in an attempt to differentiate between the two species by studying mtDNA control regions in populations in the USA and Brazil. Their phylogenetic and population genetic analyses showed high similarity between the two species, which is compatible with the single species hypothesis. Their group later studied a larger set of samples (414 individuals) using the same mtDNA region [31]. The resulting phylogenetic tree and haplotype network did not indicate phylogeographic substructuring between the two species, but rather intense haplotype sharing. In further studies, they expanded the number of samples and added nuclear genes to the studied genetic regions [32]. They found that, in Brazil, *L. purpureus* had high levels of genetic diversity distributed homogeneously throughout the analyzed geographic region, implying high effective population size and a large dispersal of individuals.

In a very recent study interrogating nuclear and mtDNA regions, da Silva et al. [47] found significant numbers of haplotypes shared between the two species, particularly in the analyzed nuclear regions. The molecular delimitation of the species supported limited discrimination between *L. purpureus* and *L. campechanus* as distinct evolutionary units. However, it did identify a substantial north–south unidirectional gene flow, suggesting that introgression was responsible for the presence of shared haplotypes. In a more comprehensive study, Pedraza-Marron et al. [62] studied samples from different populations in the USA, Mexico, Caribbean and Brazil, interrogating mtDNA regions and thousands of nuclear SNPs genotyped by RADseq. They found that the mtDNA regions failed to delimit the nominal species as distinct haplo-groups, which agrees with previous studies [29,31]. On the contrary, even though they did find evidence of introgression in neighboring populations in northern South America, they suggested that the genomic analyses strongly support the isolation and differentiation of these species, and that the northern and southern red snapper populations should be treated as distinct taxonomic entities.

The apparent contradiction between mtDNA gene-based studies and the present genomic data-based study, as well as the sharing of haplotypes among populations separated by large geographic distances, can be understood as evidence supporting strong gene flow between the two species. We believe this is evidence that the two species are going through a process of recent speciation caused by geographic isolation and environmental adaptation, and that reproductive barriers have not yet been established. The fact that the Tajima test indicated non-neutral changes in some of the populations suggests that the studied populations are subject to natural forces that are driving greater genetic variation and differentiation, which may explain previous findings based on nuclear data [62].

Generally, the high degree of mixing between the northern and southern red snapper populations in the western Atlantic is due to egg and larvae introgression. This is at the mercy of the ocean currents, suggesting that gene flow patterns among the studied populations are influenced by oceanic currents that flow from Brazil towards the Caribbean, and from the Caribbean into the Gulf of Mexico [61,63]. For example, a virtual larval tracking model for *L. analis* suggested that the marine areas of the Mesoamerican Reef are closely connected to the Gulf of Mexico through a south-to-north ocean current [64]. Nonetheless, apparent north-to-south gene flow between *L. campechanus* and *L. purpureus* has also been reported [62], which would explain the broad distribution and low genetic differentiation between red snapper populations in the western Atlantic.

The greater genetic differentiation observed between the Gulf of Mexico and Brazil populations is probably caused by an isolation-by-distance effect. A common observation is that genetic divergence between populations increases as geographic distance increases; this is the expected isolation-by-distance pattern if gene flow and genetic drift are roughly in balance [65]. Therefore, the diversification pattern observed between the Gulf of Mexico and Brazil populations may be manifesting an incipient speciation process.

## 5. Conclusions

Investigating speciation among marine organisms is complex. The combination of otolith morphometrics and genetic analyses used in the present study provided salient insights into the speciation processes. The new data generated here confirm that, in the western Atlantic, the two studied red snapper taxonomic entities *L. campechanus* and *L. purpureus* exhibit some otolith shape and genetic differentiation between populations in the Gulf of Mexico, the Caribbean, and the southwestern Atlantic, but not enough to consider them as two distinct species. It is more probable that they are in a recent speciation process generated by isolation-by-distance and adaptation to different environmental conditions.

**Supplementary Materials:** The following are available online at https://www.mdpi.com/article/10.3390/fishes7020085/s1. Table S1: Accession numbers of the sequences of the red snapper Lutjanus purpureus/Lutjanus campechanus generated in this study. Table S2: Accession numbers of the sequences for red snapper Lutjanus purpureus and Lutjanus campechanus published in GenBank used in the analyses. Table S3: Accession numbers of the sequences of Lutjanus peru, Lutjanus synagris and Lutjanus guttatus used to calculate the consensus sequences for the phylogenetic analysis. Table S4: Measures of mitochondrial DNA based on the D-loop region from red snapper captured in the western Atlantic. Table S5: Measures of nuclear DNA based on intron 8 (FADS2) from red snapper captured in the western Atlantic. Table S6: Pairwise FST values (above the diagonal) and the associated p value (below the diagonal) from the comparison between populations using the mtDNA D-loop sequences. Table S7: Pairwise FST values (above the diagonal) and the associated *p* value (below the diagonal) from the comparison between populations of red snapper using intron 8 (FADS2) sequences.

**Author Contributions:** A.M.-R., conceptualization, methodology, software, data analysis, writing—original draft; X.R., conceptualization, data analysis, writing—review and editing; G.G.-C., conceptualization, data analysis, review and editing; S.A.-R., sampling, data analysis; M.d.L.J.-B., data analysis, review and editing; J.L.M.-M.: data curation, methodology, review and editing; H.R., methodology, validation, review and editing; T.B., data analysis, review and editing; M.D.D., conceptualization, data analysis, writing—original draft, review and editing. All authors have read and agreed to the published version of the manuscript.

**Funding:** This research was funded by CONACyT postgraduate scholarship No. 730163; the Universidad Veracruzana supported the publication process.

**Institutional Review Board Statement:** All samples were obtained from dead individuals caught by the commercial multi-species artisanal fleet and following the recommendations of the Ethics Committee of the Academic Group: Management and Conservation of Aquatic Resources of the Instituto de Ciencias Marinas y Pesquerias of the Universidad Veracruzana, Boca del Rio, Veracruz, Mexico.

**Data Availability Statement:** All the supporting data can be obtained by request from the corresponding author. Sequence data was submitted to GenBank and can be accessed directly in the NCBI web page. See Table S1.

**Acknowledgments:** The authors thank the Laboratorio de Evaluación de Recursos Pesqueros (Venezuela) for technical support, Project CONACyT LAB-2009-123913 for acquisition of MEB otolith images, Universidad Veracruzana for manuscript translation and ICIMAP and Centro de Bioingenierías del Tecnológico de Monterrey for genetic analysis.

**Conflicts of Interest:** This manuscript was read and approved by all the authors listed and has not been submitted for publication, in whole or in part, to any other journal. The authors declare no conflict of interest.

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
