# Peer review of "Assessing the Speciation of Lutjanus campechanus and Lutjanus purpureus through Otolith Shape and Genetic Analyses"

_fishes, doi:10.3390/fishes7020085_

Round 1

Reviewer 1 Report

The manuscript Assessing speciation of Lutjanus campechanus and Lutjanus purpureus through otolith shape and genetic analyses by Marval-Rodríguez et al. is very well prepared. I am familiar with the genetic methods, otoliths are not the main focus in my studies, but I have some experience too. I carefully read the paper, checked the data and their analysis, and on the methods, data presentation and conclusions based on the results I do not have any comments, all seems to me to be done correctly.

The only issue is that the results are contradicting partly the recent work of Pedraza-Marron et al. (2019). These authors used genome wide data (SNP from RAD sequencing) to evaluate the genetic structure of red snappers in WA and their results clearly suggest strong genetic differentiation of the two putative species (L. campechanus and L. purpureus), and they concluded they represent two evolutionary entities/species based on the genomic data. Genomic data are much more informative then mtDNA markers, which in recently evolved (or evolving) species often does not show any genetic differentiation due to numerous reasons (discussed in the MS). The low genetic differentiation (admixing) was repeatedly observed on red snappers (references in the MS). Otoliths provided only limited information on population diversification too. However, in the discussion is only marginally mentioned this disagreement. In lines 479-482 is included only information about observed pattern of unidirectional limited gen-flow in Pedraza-Marron et al. (2019). Include in the discussion also their main finding (strong genetic differentiation) and explain why there is the difference in observed patter in your data and their results.

At the end of the discussion (lines 490-494) you in fact suggest the two entities are speciating recently, which is in line of Pedraza-Marron et al. (2019).

Author Response

Thanks for the suggestion. We added three paragraphs to further compared our results to previous studies, including the recent study by Pedraza-Marron et al. (2019). We point out the contradicting results from the nuclear and mtDNA data and elaborate further on the significance of these findings.

Reviewer 2 Report

The manuscript contains detailed research on the morphometric and genetic investigation of taxonomic identification of two Lutjanus species: L. campechanus and L. purpureus. The subject of the paper is worthy of investigation.

Methods and experimental design are seemed to be correct. However, some revision is needed in the manuscript.

Introduction should be extended with a short paragraph on the application of otolith morphometrics in taxonomic identification and its usage in the Lutjanidae family.

Material and methods:

Sample collection: although the sample sizes could be found in tables, please include the sample sizes collected for genetic analysis for each population.

The correlation analysis is not included in the methodical description of the genetic analysis.

Results:

Line 293- 294. “Tajima’s test revealed significant differences … between the last two populations”- what does mean? which populations? Tajima’s D test indicates neutrality, the presence or absence of selection based on the diversity measurements of a marker in a population.

The FST values on a population level – AMOVA shows that variation among populations is not significant. In my opinion the presentation of the pairwise FST values in the text is not required, as they show that only regional differences were significant. Even the low sample numbers in some of the populations could influence the FST value (as it could be seen from the non-significant negative values) The tables should be presented as supplementary material.

For the investigation of the association between the genetic and geographic distances the correct method is the Mantel test.

Discussion:

Line 421-426 – Please rephrase the text. The content is misleading as it suggests that the two examined genes has direct effect on the otolith development. However, FADS2 is included in the fatty acid metabolism, mtDNA CR is a non-coding DNA. Besides, it does not mean that the differences have no genetic background, several genes are identified as candidates in the otolith development.

Minor problems:

Line 89- Scientific names are italic

Table 5 and Table 7 – what does mean + symbol?

Figure 2 Please define Lt

Author Response

The manuscript contains detailed research on the morphometric and genetic investigation of taxonomic identification of two Lutjanus species: L. campechanus and L. purpureus. The subject of the paper is worthy of investigation.

Methods and experimental design are seemed to be correct. However, some revision is needed in the manuscript.

Introduction should be extended with a short paragraph on the application of otolith morphometrics in taxonomic identification and its usage in the Lutjanidae family.

R: A paragraph in the introduction has been added with the requested information on otoliths and Lutjanidae family

“Otolith shape analysis has been used for more than 20 years as an objective method to identify species, to discriminate fish stocks, in systemic and taxonomic studies, for aging fish, in fish auditory neuroscience and to study the ecomorphological patterns of fish (13,14,15,16,17,18,19,20,21). For the Lutjanidae family, in particular, otolith shape analysis has been used to determine the environmental and genetic influences in otolith morphology, to age juvenile red snapper, for stock discrimination, to identify closely related species and to analyze morphometric relationships between two species (22,23,24,25,26).”

Material and methods:

Sample collection: although the sample sizes could be found in tables, please include the sample sizes collected for genetic analysis for each population.

R: We noted in tables 1 and 2 the number of samples that were processed and analyzed in our study. This is also stated in Supplemental table 1

The correlation analysis is not included in the methodical description of the genetic analysis.

R: We added the information on the Mantel test as requested below.

Results:

Line 293- 294. “Tajima’s test revealed significant differences … between the last two populations”- what does mean? which populations? Tajima’s D test indicates neutrality, the presence or absence of selection based on the diversity measurements of a marker in a population.

R: Thank you for this observation, indeed the phrase was not worded correctly. This and a previous sentence were reworded to make the findings clear as follows:

Section 3.2.1: “Tajima's test for the mtDNA-CR region revealed that the changes found were significantly different from random changes (non-neutral)”

“Tajima's test revealed that changes found within populations of South Veracruz, Rio Caribe and Margarita, as well as between the Venezuelan populations were non-neutral (Table 5), suggesting the action of selection or other forces driving the variability in a directed way”

The FST values on a population level – AMOVA shows that variation among populations is not significant. In my opinion the presentation of the pairwise FST values in the text is not required, as they show that only regional differences were significant. Even the low sample numbers in some of the populations could influence the FST value (as it could be seen from the non-significant negative values). The tables should be presented as supplementary material.

R: Tables were changed as supplementary tables 5 and 6.

For the investigation of the association between the genetic and geographic distances the correct method is the Mantel test.

R: We carry out a Mantel test to evaluate the correlation between geographic and genetic distances. We added the text in materials and methods as well as in the results section.

“In addition, to test for a correlation between genetic and geographic distances among the populations, we carried out a Mantel tests using the geographic distance calculated from Google Earth (drawing straight lines between the sampling areas) and pairwise FST (obtained from the AMOVA). For this, we used GenAlEx, version 6.51b2 (50)”

“The Mantel tests revealed a significant correlation between geographic and genetic distance among the populations (rM = 0.571; P = 0.010; Figure 4), which suggest little gene flow among regions.”

Discussion:

Line 421-426 – Please rephrase the text. The content is misleading as it suggests that the two examined genes has direct effect on the otolith development. However, FADS2 is included in the fatty acid metabolism, mtDNA CR is a non-coding DNA. Besides, it does not mean that the differences have no genetic background, several genes are identified as candidates in the otolith development.

R: Thanks for pointing this out. We change the text to eliminated the sentence.

Minor problems:

Line 89- Scientific names are italic

R: Corrected

Table 5 and Table 7 – what does mean + symbol?

R: This was a typo; it was changed to *

Figure 2 Please define Lt

R: Corrected

Reviewer 3 Report

Line 32. vital to improve or vital for improving.

Line 81. Figure 1. It would be more appropriate in two sampling sites would not have the same marking (CN1 - see also line 86)

Line 87. There is a dot in the subtitle.

Line 89. Names of the species should be written in italik.

Lines 95-129. This has to be reduced, it is too long and already known.

Line 132. ...established before. Delete everything in parenthesis.

Lines 131-160. Also could be shorten. This is not first time that this statistics is applied.

Lines 164-177. Just mention the paper where it has been previously used this kind of procedure for extraction. Do not describe it. The same applies for lines 179-192, and Table 1 should be omitted.

Line 200- haplotypes,

Line 207. They were searched in the GenBank using BLAST.  I presume that authors wanted to say that their sequences were depositet in GenBank, because in line 216 thay say that used other sequences.

Line 227. Same as for line 89.

Line 251. red snapper considering...

Table 2. Considering the fact that most indices did not show statistically significant difference, this table should be omitted and text will be sufficient. 

The same applies for Table 3., 4. and 5. They can also be omitted.

Line 323. There is no CN1 in the Table 8! Only CN is present, and now it is important to see the comment in line 81. We do not know what sample it is.

Lines 332. LA, MS, FL and CN are not in the Table 9!

Figure 6. This kind of tree shows very poor segregation of samples used in this research. It is not because they are very close it is because other phylogenetic analysis should be applied to have beter resolutions for obtained samples. Not all species from this family are necessary to be in the new tree.

Line 377. ostium, not in italik.

Line 387. Citation 50 is not appropriate for this statement. Find other.

Line 390. (51,52), omit number 529.

Lines 390-409. This whole paragraph should be omitted because has no relevance with obtained results.

Lines 410-412. If this statement is correct, why have the samples from Campeche been taken into analises?

Line 417. Citation 60 is not appropriate for this statement. Find other.

Lines 446-456. The whole paragraph should be omitted for it has no relevance with the obtained results.

Lines 490-494. I am not certain that obtained results support this conclusion.

Author Response

Comments and Suggestions for Authors

Line 32. vital to improve or vital for improving.

R: Corrected

Line 81. Figure 1. It would be more appropriate in two sampling sites would not have the same marking (CN1 - see also line 86)

R: Corrected

Line 87. There is a dot in the subtitle.

R: Corrected

Line 89. Names of the species should be written in italik.

R: Corrected

Lines 95-129. This has to be reduced, it is too long and already known.

R: All materials and methods for otolith shape analysis has been reduced as requested

Line 132. ...established before. Delete everything in parenthesis.

R: Corrected

Lines 131-160. Also could be shorten. This is not first time that this statistics is applied.

R: All materials and methods for otolith shape analysis has been reduced as requested

Lines 164-177. Just mention the paper where it has been previously used this kind of procedure for extraction. Do not describe it. The same applies for lines 179-192, and Table 1 should be omitted.

R: Details of the DNA extraction and amplification protocols were eliminated, as well as table 1. The text was summarized as follows:

“DNA extracting was carried out from alcohol-preserved muscle tissue using Wizard® Genomic DNA Purification Kit (Promega), following the recommendation of the manufacturer. DNA concentration and quality were measured using a NanoDrop One (Thermo Scientific) and by visual inspection of the DNA by electrophoresis. DNA extracts were stored at -20°C until use”

“The fragments were sequenced using both primers and the BigDye Terminator chemistry in the Laboratorio Nacional de Genómica para la Biodiversidad (Langebio), Irapuato, Mexico”

Line 200- haplotypes,

R: Corrected

Line 207. They were searched in the GenBank using BLAST.  I presume that authors wanted to say that their sequences were depositet in GenBank, because in line 216 thay say that used other sequences.

R: Yes, they were deposited in GenBank. However, we wanted to say that we search for published sequences in GenBank using BLAST. We changed the wording as follows:

“They were compared to sequences in GenBank using BLAST”

Line 227. Same as for line 89.

R: Corrected

Line 251. red snapper considering...

R: Corrected

Table 2. Considering the fact that most indices did not show statistically significant difference, this table should be omitted and text will be sufficient. 

R: Table 2 was eliminated as requested

The same applies for Table 3., 4. and 5. They can also be omitted.

R: Tables 4 and 5 (now tables 1 and 2) show the number of samples analyzed for each gene and is showing significant values for the Tajima Test.

Line 323. There is no CN1 in the Table 8! Only CN is present, and now it is important to see the comment in line 81. We do not know what sample it is.

R: Samples CN1 and CN2 were clustered, since in GenBank there was no information about which was what. Since they were from very close sampling sites, we believe that this will not affect the analyses.

Lines 332. LA, MS, FL and CN are not in the Table 9!

R: No published data was available for USA sites in GenBank. CN is shown in table 9.

Figure 6. This kind of tree shows very poor segregation of samples used in this research. It is not because they are very close it is because other phylogenetic analysis should be applied to have beter resolutions for obtained samples. Not all species from this family are necessary to be in the new tree.

R: The phylogenetic figure is presented to show that there is clear differentiation among L. synagris, L. peru/L. campechanus/L. purpureus and the Western Pacific members of Lutjanus, which shows statistical support >90. Even that the distances cannot be used to establish precise differentiation, it can serve to have a relative idea of distances among these three clusters. These are the only sequences available in species of Lutjanus. It really has low resolution, due to the nature of the sequence, but no other comparison can be made, and is not the objective of this study to study the phylogenetic relationship of the whole genus.

Line 377. ostium, not in italik.

R: Corrected

Line 387. Citation 50 is not appropriate for this statement. Find other.

R: Thank you for this observation, indeed the citation was modified with another more pertinent one

Line 390. (51,52), omit number 529.

R: Corrected

Lines 390-409. This whole paragraph should be omitted because has no relevance with obtained results.

R: Paragraph was edited to improve clarity. We still believe information in this paragraph is important to establish that there are environmental differences between regions and fishing areas, transcendental to our findings.

Lines 410-412. If this statement is correct, why have the samples from Campeche been taken into analyses?

R: All otoliths that could be obtained through the fishing fleets were used in this study. Unfortunately, our institutions do not have a research vessel that could be used to obtained specific samples. The results show that otoliths from Campeche discriminate form others, probably related to their smaller size compared to those from the other areas. However, since these samples provide important information for the genetic characterization, we kept them in the study. We agree with Bird et al. (1986) and Campana and Casselman (1993), cited in Hüssy (2008), that differences between age groups, particularly early in fish life seem to have otolith shape differences due to their development stage, then we thought we would likely see differences in our genetic analyses. We kept the otolith data from Campeche, but pointing out that their differences may be associated to their developmental stage.

Line 417. Citation 60 is not appropriate for this statement. Find other.

R: Thank you for this observation, indeed the citation was modified with another more pertinent one

Lines 446-456. The whole paragraph should be omitted for it has no relevance with the obtained results.

R: The text was reduced to include only the most relevant discussion about the results.

Lines 490-494. I am not certain that obtained results support this conclusion.

R: The conclusion was changed.

Round 2

Reviewer 3 Report

Page 2. New paragraph - ...to determine age of juvenile red snapper,...

Page 4. Molecular analysis - ...according to previously published studies (46, 46). If there are more studies use different numbers. If there is only one study them marked it properly

Tables 4 and 5 (now table 1 and 2). It is accurate that these tables show samples analyzed for each gene and that they are showing significant values for Tajima Test. However, only four values in table 1 and three values in table 2 are statistically significant. For the readibility of the MS it is better to include this statistically significant values in the text, and put these tables in supplement material. 

Figure 6. I agree with the authors that the objective of this study was not the phylogenetic relationship of the whole genus. That is main reason to make remark on this figure. Also, the objective of this study was not the differentiation among L. synagris, L. peru/L. campechanus/L. purpureus and the Western Pacific members of genus Lutjanus. The objective of this study was differentiation among  L. campechanus and L. purpureus and this figure is not informative on this objective. It only helps to see where these two species stand in the genus, not how they are compared (separated or not). I suggest other phylogenetic tree to be made, to see proposed objective of this study. If not, separation on the basis of genetic analysis can not be made, as well as conclusion based on this.

Page 12. "Our result do not show statistical differences between red snapers in the Gulf of Mexico and Caribbean basin supporting the theory that they belong to one species, if we consider that otolith shape is species-specific and can be used to differentiate species (58).

In this citation (58), authors could make identification of only six of the 41 species analysed, emphasizeing the complexity of relationship between otholith morphometry and environmental or genetic factors (among genera and species). Leading that the statement of the authors is somewhat biased in my opinion.

Also, if the authors still want to keep part of the paragraph that was suggested to be omitted, they should know that this part contains data about ecological and environmental conditions (temperature, salinity and sediment) which do not favour their theory about morphometric plasticity being species-specific. Also, these considred parameters (temp., sal., sediment) authors did not have in their study (for caught L. campechanus and L. purpureus), making it imposible to compare or disscuss.

Page 13. Second sentence in 4.2. Genetic differentiation is broken in two.

Page 13. Last pragraph that was suggested to be omitted was not. Also it was not reduced and it is almost the same as in the first version of the MS. This was suggested due to the irrelevant literature cited .Unless the authors give relevant literature (on the species investigated in this study, not other species elsewhere). This whole paragrapf should be omitted.

Page 14. End of first new paragraph - ...and large of dispersal of the individuals. Omitt first "of".  Further mistake - (,47), delete comma.

Page 14. The whole new text on genetic analyses deepens more problematic on this subject. In concordance with this, my proposal of more detailed phylogenetic tree is essential. That is the reason for authors that new analysis has to be made to show more light on separation or no separation of these two species. I found genetic analysis done in this paper not sufficient enough.

Author Response

Page 2. New paragraph - ...to determine age of juvenile red snapper,...

  1. R) = Done.

Page 4. Molecular analysis - ...according to previously published studies (46, 46). If there are more studies use different numbers. If there is only one study them marked it properly

  1. R) = Corrected.

Tables 4 and 5 (now table 1 and 2). It is accurate that these tables show samples analyzed for each gene and that they are showing significant values for Tajima Test. However, only four values in table 1 and three values in table 2 are statistically significant. For the readibility of the MS it is better to include this statistically significant values in the text, and put these tables in supplement material.

  1. R) = Reviewer 2 ask us to show the number of individuals analyzed in each population, thus it is contradictory with this reviewer. Please advise what would be the best option.

Figure 6. I agree with the authors that the objective of this study was not the phylogenetic relationship of the whole genus. That is main reason to make remark on this figure. Also, the objective of this study was not the differentiation among L. synagris, L. peru/L. campechanus/L. purpureus and the Western Pacific members of genus Lutjanus. The objective of this study was differentiation among L. campechanus and L. purpureus and this figure is not informative on this objective. It only helps to see where these two species stand in the genus, not how they are compared (separated or not). I suggest other phylogenetic tree to be made, to see proposed objective of this study. If not, separation on the basis of genetic analysis can not be made, as well as conclusion based on this.

  1. R) = Since the objective of this study was not the phylogenetic relationship of the whole genus, we eliminated all the sequences from the other Lutjanus species, except for synagris, L. guttatus and L. peru, which we used as outgroups to study the phylogenetic relationships among the populations of L. campechanus and L. purpureus. We also used a Bayesian analysis because is more robust than the ML method. This created changes in the scale and the resolution of the phylogenetic differentiation among the studied populations. Now, the analysis shows a separation between the two clades with high statistical support with monophyly in each group. We thanks the reviewer because this change made more clear the separation of the “species”.

Page 12. "Our result do not show statistical differences between red snapers in the Gulf of Mexico and Caribbean basin supporting the theory that they belong to one species, if we consider that otolith shape is species-specific and can be used to differentiate species (58). In this citation (58), authors could make identification of only six of the 41 species analysed, emphasizeing the complexity of relationship between otholith morphometry and environmental or genetic factors (among genera and species). Leading that the statement of the authors is somewhat biased in my opinion. Also, if the authors still want to keep part of the paragraph that was suggested to be omitted, they should know that this part contains data about ecological and environmental conditions (temperature, salinity and sediment) which do not favour their theory about morphometric plasticity being species-specific. Also, these considred parameters (temp., sal., sediment) authors did not have in their study (for caught L. campechanus and L. purpureus), making it imposible to compare or disscuss.

  1. R) = Rewritten

Page 13. Second sentence in 4.2. Genetic differentiation is broken in two.

  1. R) = Corrected.

Page 13. Last pragraph that was suggested to be omitted was not. Also it was not reduced and it is almost the same as in the first version of the MS. This was suggested due to the irrelevant literature cited .Unless the authors give relevant literature (on the species investigated in this study, not other species elsewhere). This whole paragrapf should be omitted.

  1. R) = We eliminated 11/18 lines of this paragraph, but consider that the first part of the paragraph is important to explain why there are many shared haplotypes among the studied populations.

Page 14. End of first new paragraph - ...and large of dispersal of the individuals. Omitt first "of".  Further mistake - (,47), delete comma.

  1. R) = Corrected.

Page 14. The whole new text on genetic analyses deepens more problematic on this subject. In concordance with this, my proposal of more detailed phylogenetic tree is essential. That is the reason for authors that new analysis has to be made to show more light on separation or no separation of these two species. I found genetic analysis done in this paper not sufficient enough.

  1. R) =We think that the new analysis shows more clear separation, which further agrees with the results from Pedraza-Marron et al. (2019) and da Silva et al. 2020.